# A General Theory of Equivariant CNNs on Homogeneous Spaces

**Taco S. Cohen**
Qualcomm AI Research*
Qualcomm Technologies Netherlands B.V.
`tacos@qti.qualcomm.com`

**Mario Geiger**
PCSL Research Group
EPFL
`mario.geiger@epfl.ch`

**Maurice Weiler**
QUVA Lab
U. of Amsterdam
`m.weiler@uva.nl`

## Abstract

We present a general theory of Group equivariant Convolutional Neural Networks (G-CNNs) on homogeneous spaces such as Euclidean space and the sphere. Feature maps in these networks represent fields on a homogeneous base space, and layers are equivariant maps between spaces of fields. The theory enables a systematic classification of all existing G-CNNs in terms of their symmetry group, base space, and field type. We also consider a fundamental question: what is the most general kind of equivariant linear map between feature spaces (fields) of given types? Following Mackey, we show that such maps correspond one-to-one with convolutions using equivariant kernels, and characterize the space of such kernels.

## 1 Introduction

Through the use of convolution layers, Convolutional Neural Networks (CNNs) have a built-in understanding of *locality* and *translational symmetry* that is inherent in many learning problems. Because convolutions are *translation equivariant* (a shift of the input leads to a shift of the output), convolution layers preserve the translation symmetry. This is important, because it means that further layers of the network can also exploit the symmetry.

Motivated by the success of CNNs, many researchers have worked on generalizations, leading to a growing body of work on *Group equivariant CNNs* (G-CNNs) for signals on Euclidean space and the sphere [1–7] as well as graphs [8, 9]. With the proliferation of equivariant network layers, it has become difficult to see the relations between the various approaches. Furthermore, when faced with a new modality (diffusion tensor MRI, say), it may not be immediately obvious how to create an equivariant network for it, or whether a given kind of equivariant layer is the most general one.

In this paper we present a general theory of homogeneous G-CNNs. Feature spaces are modelled as spaces of fields on a homogeneous space. They are characterized by a group of symmetries $G$, a subgroup $H \leq G$ that together with $G$ determines a homogeneous space $B \simeq G/H$, and a representation $\rho$ of $H$ that determines the type of field (vector, tensor, etc.). Related work is classified by $(G, H, \rho)$. The main theorems say that equivariant linear maps between fields over $B$ can be written as convolutions with an equivariant kernel, and that the space of equivariant kernels can be realized in three equivalent ways. We will assume some familiarity with groups, cosets, quotients, representations and related notions (see Appendix A).

This paper does not contain truly new mathematics (in the sense that a professional mathematician with expertise in the relevant subjects would not be surprised by our results), but instead provides a new formalism for the study of equivariant convolutional networks. This formalism turns out to be a remarkably good fit for describing real-world G-CNNs. Moreover, by describing G-CNNs in a language used throughout modern physics and mathematics (fields, fiber bundles, etc.), it becomes possible to apply knowledge gained over many decades in those domains to machine learning.

## 1.1 Overview of the Theory

This paper has two main parts. First, in Sec. 2, we introduce a mathematical model for convolutional feature spaces. The basic idea is that feature maps represent fields over a homogeneous space. As it turns out, defining the notion of a field is quite a bit of work. So in order to motivate the introduction of each of the required concepts, we will in this section provide an overview of the relevant concepts and their relations, using the example of a Spherical CNN with vector field feature maps.

The second part of this paper (Section 3) is about maps between the feature spaces. We require these to be equivariant, and focus in particular on the linear layers. The main theorems (3.1–3.4) show that linear equivariant maps between the feature spaces are in one-to-one correspondence with equivariant convolution kernels (i.e. *convolution is all you need*), and that the space of equivariant kernels can be realized as a space of matrix-valued functions on a group, coset space, or double coset space, subject to linear constraints.

In order to specify a convolutional feature space, we need to specify two things: a homogeneous space $B$ over which the field is defined, and the type of field (e.g. vector field, tensor field, etc.). A homogeneous space for a group $G$ is a space $B$ where for any two $x, y \in B$ there is a transformation $g \in G$ that relates them via $gx = y$. Here we consider the example of a vector field on the sphere $B = S^2$ with symmetry group $G = \mathrm{SO}(3)$, the group of 3D rotations. The sphere is a homogeneous space for $\mathrm{SO}(3)$ because we can map any point on the sphere to any other via a rotation.

Formally, a field is defined as a *section* of a *vector bundle associated to a principal bundle*. In order to understand what this means, we must first know what a *fiber bundle* is (Sec. 2.1), and understand how the group $G$ can be viewed as a *principal bundle* (Sec. 2.2). Briefly, a fiber bundle formalizes the idea of parameterizing a set of identical spaces called *fibers* by another space called the *base space*.

The first way in which fiber bundles play a role in the theory is that the action of $G$ on $B$ allows us to think of $G$ as a "bundle of groups" or *principal bundle*. Roughly speaking, this works as follows: if we fix an origin $o \in B$, we can consider the *stabilizer subgroup* $H \leq G$ of transformations that leave $o$ unchanged: $H = \{g \in G \mid go = o\}$. For example, on the sphere the stabilizer is $\mathrm{SO}(2)$, the group of rotations around the axis through $o$ (e.g. the north pole). As we will see in Section 2.2, this allows us to view $G$ as a bundle with *base space* $B \simeq G/H$ and a *fiber* $H$. This is shown for the sphere in Fig. 1 (cartoon). In this case, we can think of $\mathrm{SO}(3)$ as a bundle of circles ($H = \mathrm{SO}(2)$) over the sphere, which itself is the quotient $S^2 \simeq \mathrm{SO}(3)/\mathrm{SO}(2)$.

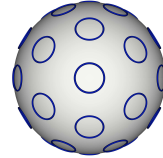

Figure 1: $\mathrm{SO}(3)$ as a principal $\mathrm{SO}(2)$ bundle over $S^2$.

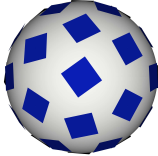

Figure 2: Tangent bundle of $S^2$.

To define the *associated bundle* (Sec. 2.3) we take the principal bundle $G$ and replace the fiber $H$ by a vector space $V$ on which $H$ acts linearly via a *group representation* $\rho$. This yields a vector bundle with the same base space $B$ and a new fiber $V$. For example, the tangent bundle of $S^2$ (Fig. 2) is obtained by replacing the circular $\mathrm{SO}(2)$ fibers in Fig. 1 by 2D planes. Under the action of $H = \mathrm{SO}(2)$, a tangent vector at the north pole is rotated (even though the north pole itself is fixed by $\mathrm{SO}(2)$), so we let $\rho(h)$ be a $2 \times 2$ rotation matrix. In a general convolutional feature space with $n$ channels, $V$ would be an $n$-dimensional vector space. Finally, fields are defined as *sections* of this bundle, i.e. an assignment to each point $x$ of an element in the fiber over $x$ (see Fig. 3).

Having defined the feature space, we need to specify how it transforms (e.g. say how a vector field on $S^2$ is rotated). The natural way to transform a $\rho$-field is via the *induced representation* $\pi = \mathrm{Ind}_H^G \rho$ of $G$ (Section 2.4), which combines the action of $G$ on the base space $B$ and the action of $\rho$ on the fiber $V$ to produce an action on sections of the associated bundle (See Figure 3). Finally, having defined the feature spaces and their transformation laws, we can study equivariant linear maps between them (Section 3). In Sec. 4–6 we cover implementation aspects, related work, and concrete examples, respectively.

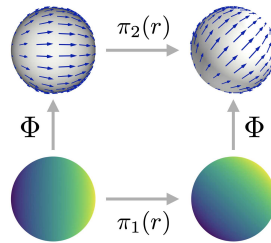

Figure 3: $\Phi$ maps scalar fields to vector fields, and is equivariant to the induced representation $\pi_i = \mathrm{Ind}_{\mathrm{SO}(2)}^{\mathrm{SO}(3)} \rho_i$.

## 2 Convolutional Feature Spaces

### 2.1 Fiber Bundles

Intuitively, a fiber bundle is a parameterization of a set of isomorphic spaces (the fibers) by another space (the base). For example, we can think of a feature space in a classical CNN as a set of vector spaces $V_x \simeq \mathbb{R}^n$ ($n$ being the number of channels), one per position $x$ in the plane [2]. This is an example of a *trivial bundle*, because it is simply the Cartesian product of the plane and $\mathbb{R}^n$. General fiber bundles are only *locally trivial*, meaning that they locally look like a product while having a different global topological structure.

The simplest example of a non-trivial bundle is the Mobius strip, which locally looks like a product of the circle (the base) with a line segment (the fiber), but is globally distinct from a cylinder (see Fig. 4). A more practically relevant example is given by the tangent bundle of the sphere (Fig. 2), which has as base space $S^2$ and fibers that look like $\mathbb{R}^2$, but is topologically distinct from $S^2 \times \mathbb{R}^2$ as a bundle.

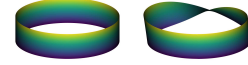

Figure 4: Cylinder and Möbius strip

Formally, a bundle consists of topological spaces $E$ (total space), $B$ (base space), $F$ (canonical fiber), and a projection map $p : E \to B$, satisfying a local triviality condition. Basically, this condition says that locally, the bundle looks like a product $U \times F$ of a piece $U \subseteq B$ of the base space, and $F$ the canonical fiber. Formally, the condition is that for every $a \in E$, there is an open neighbourhood $U \subseteq B$ of $p(a)$ and a homeomorphism $\varphi : p^{-1}(U) \to U \times F$ so that the map $p^{-1}(U) \xrightarrow{\varphi} U \times F \xrightarrow{\text{proj}_1} U$ agrees with $p : p^{-1}(U) \to U$ (where $\text{proj}_1(u, f) = u$). The homeomorphism $\varphi$ is said to locally trivialize the bundle above the trivializing neighbourhood $U$.

Considering that for any $x \in U$ the preimage $\text{proj}_1^{-1}(x)$ is $F$, and $\varphi$ is a homeomorphism, we see that the preimage $F_x = p^{-1}(x)$ for $x \in B$ is also homeomorphic to $F$. Thus, we call $F_x$ the fiber over $x$, and see that all fibers are homeomorphic. Knowing this, we can denote a bundle by its projection map $p : E \to B$, leaving the canonical fiber $F$ implicit.

Various more refined notions of fiber bundle exist, each corresponding to a different kind of fiber. In this paper we will work with principal bundles (bundles of groups) and vector bundles (bundles of vector spaces).

A *section* $s$ of a fiber bundle is an assignment to each $x \in B$ of an element $s(x) \in F_x$. Formally, it is a map $s : B \to E$ that satisfies $p \circ s = \text{id}_B$. If the bundle is trivial, a section is equivalent to a function $f : B \to F$, but for a non-trivial bundle we cannot continuously align all the fibers simultaneously, and so we must keep each $s(x)$ in its own fiber $F_x$. Nevertheless, on a trivializing neighbourhood $U \subseteq B$, we can describe the section as a function $s_U : U \to F$, by setting $\varphi(s(x)) = (x, s_U(x))$.

### 2.2 $G$ as a Principal $H$-Bundle

Recall (Sec. 1.1) that with every feature space of a G-CNN is associated a homogeneous space $B$ (e.g. the sphere, projective space, hyperbolic space, Grassmann & Stiefel manifolds, etc.), and recall further that such a space has a stabilizer subgroup $H = \{g \in G \mid go = o\}$ (this group being independent of origin $o$ up to isomorphism). As discussed in Appendix A, the cosets $gH$ of $H$ (e.g. the circles in Fig. 1) partition $G$, and the set of cosets, denoted $G/H$ (e.g. the sphere in Fig. 1), can be identified with $B$ (up to a choice of origin).

It is this partitioning of $G$ into cosets that induces a special kind of bundle structure on $G$. The projection map that defines the bundle structure sends an element $g \in G$ to the coset $gH$ it belongs to. Thus, it is a map $p : G \to G/H$, and we have a bundle with total space $G$, base space $G/H$ and canonical fiber $H$. Intuitively, this allows us to think of $G$ as a base space $G/H$ with a copy of $H$ attached at each point $x \in G/H$. The copies of $H$ are glued together in a potentially twisted manner.

This bundle is called a *principal $H$-bundle*, because we have a transitive and fixed-point free group action $G \times H \to G$ that preserves the fibers. This action is given by right multiplication, $g \mapsto gh$, which preserves fibers because $p(gh) = ghH = gH = p(g)$. That is, by right-multiplying an element $g \in G$ by $h \in H$, we get an element $gh$ that is in general different from $g$ but is still within the same coset (i.e. fiber). That the action is transitive and free on cosets follows immediately from the group axioms.

One can think of a principal bundle as a bundle of generalized frames or *gauges* relative to which geometrical quantities can be expressed numerically. Under this interpretation the fiber at $x$ is a space of generalized frames, and the action by $H$ is a change of frame. For instance, each point on the circles in Fig. 1 can be identified with a right-handed orthogonal frame, and the action of $SO(2)$ corresponds to a rotation of this frame. The group $H$ may also include internal symmetries, such as color space rotations, which do not relate in any way to the spatial dimensions of $B$.

In order to numerically represent a field on some neighbourhood $U \subseteq G/H$, we need to choose a frame for each $x \in U$ in a continuous manner. This is formalized as a section of the principal bundle. Recall that a section of $p : G \to G/H$ is a map $s : G/H \to G$ that satisfies $p \circ s = \mathrm{id}_{G/H}$. Since $p$ projects $g$ to its coset $gH$, the section chooses a representative $s(gH) \in gH$ for each coset $gH$. Non-trivial principal bundles do not have continuous global sections, but we can always use a local section on $U \subseteq G/H$, and represent a field on overlapping local patches covering $G/H$.

Aside from the right action of $H$, which turns $G$ into a principal $H$-bundle, we also have a left action of $G$ on itself, as well as an action of $G$ on the base space $G/H$. In general, the action of $G$ on $G/H$ does not agree with the action on $G$, in that $gs(x) \neq s(gx)$, because the action on $G$ includes a twist of the fiber. This twist is described by the function $\mathrm{h} : G/H \times G \to H$ defined by $gs(x) = s(gx)\mathrm{h}(x, g)$ (whenever both $s(x)$ and $s(gx)$ are defined). This function will be used in various calculations below. We note for the interested reader that $\mathrm{h}$ satisfies the cocycle condition $\mathrm{h}(x, g_1 g_2) = \mathrm{h}(g_2 x, g_1)\mathrm{h}(x, g_2)$.

## 2.3 The Associated Vector Bundle

Feature spaces are defined as spaces of sections of the associated vector bundle, which we will now define. In physics, a section of an associated bundle is simply called a field.

To define the associated vector bundle, we start with the principal $H$-bundle $G \xrightarrow{p} G/H$, and essentially replace the fibers (cosets) by vector spaces $V$. The space $V \simeq \mathbb{R}^n$ carries a group representation $\rho$ of $H$ that describes the transformation behaviour of the feature vectors in $V$ under a change of frame. These features could for instance transform as a scalar, a vector, a tensor, or some other geometrical quantity [2, 6, 8]. Figure 3 shows an example of a vector field ($\rho(h)$ being a $2 \times 2$ rotation matrix in this case) and a scalar field ($\rho(h) = 1$).

The first step in constructing the associated vector bundle is to take the product $G \times V$. In the context of representation learning, we can think of an element $(g, v)$ of $G \times V$ as a feature vector $v \in V$ and an associated pose variable $g \in G$ that describes how the feature detector was steered to obtain $v$. For instance, in a Spherical CNN [10] one would rotate a filter bank by $g \in SO(3)$ and match it with the input to obtain $v$. If we apply a transformation $h \in H$ to $g$ and simultaneously apply its inverse to $v$, we get an equivalent element $(gh, \rho(h^{-1})v)$. In a Spherical CNN, this would correspond to a change in orientation of the filters by $h \in SO(2)$.

So in order to create the associated bundle, we take the quotient of the product $G \times V$ by this action: $A = G \times_\rho V = (G \times V)/H$. In other words, the elements of $A$ are orbits, defined as $[g, v] = \{(gh, \rho(h^{-1})v) \mid h \in H\}$. The projection $p_A : A \to G/H$ is defined as $p_A([g, v]) = gH$. One may check that this is well defined, i.e. independent of the orbit representative $g$ of $[g, v] = [gh, \rho(h^{-1})v]$. Thus, the associated bundle has base $G/H$ and fiber $V$, meaning that locally it looks like $G/H \times V$. We note that the associated bundle construction works for any principal $H$-bundle, nog just $p : G \to G/H$, which suggests a direction for further generalization [11].

A field ("stack of feature maps") is a section of the associated bundle, meaning that it is a map $s : G/H \to A$ such that $\pi_\rho \circ s = \mathrm{id}_{G/H}$. We will refer to the space of sections of the associated vector bundle as $\mathcal{I}$. Concretely, we have two ways to encode a section: as functions $f : G \to V$ subject to a constraint, and as local functions from $U \subseteq G/H$ to $V$. We will now define both.

### 2.3.1 Sections as Mackey Functions

The construction of the associated bundle as a product $G \times V$ subject to an equivalence relation suggests a way to describe sections concretely: a section can be represented by a function $f : G \to V$ subject to the equivariance condition

$$f(gh) = \rho(h^{-1})f(g). \tag{1}$$

Such functions are called Mackey functions. They provide a redundant encoding of a section of $A$, by encoding the value of the section relative to any choice of frame / section of the principal bundle simultaneously, with the equivariance constraint ensuring consistency.

A linear combination of Mackey functions is a Mackey function, so they form a vector space, which we will refer to as $\mathcal{I}_G$. Mackey functions are easy to work with because they allow a concrete and global description of a field, but their redundancy makes them unsuitable for computer implementation.

### 2.3.2 Local Sections as Functions on $G/H$

The associated bundle has base $G/H$ and fiber $V$, so locally, we can describe a section as an unconstrained function $f : U \to V$ where $U \subseteq G/H$ is a trivializing neighbourhood (see Sec. 2.1). We refer to the space of such sections as $\mathcal{I}_C$. Given a local section $f \in \mathcal{I}_C$, we can encode it as a Mackey function through the following lifting isomorphism $\Lambda : \mathcal{I}_C \to \mathcal{I}_G$:

$$
\begin{aligned}
[\Lambda f](g) &= \rho(\mathrm{h}(g)^{-1})f(gH), \\
[\Lambda^{-1}f'](x) &= f'(s(x)),
\end{aligned}
\tag{2}
$$

where $\mathrm{h}(g) = \mathrm{h}(H, g) = s(gH)^{-1}g \in H$ and $s(x) \in G$ is a coset representative for $x \in G/H$. This map is analogous to the lifting defined by [12] for scalar fields (i.e. $\rho(h) = I$), and can be defined more generally for any principal / associated bundle [13].

## 2.4 The Induced Representation

The induced representation $\pi = \mathrm{Ind}_H^G \rho$ describes the action of $G$ on fields. In $\mathcal{I}_G$, it is defined as:

$$
[\pi_G(g)f](k) = f(g^{-1}k).
\tag{3}
$$

In $\mathcal{I}_C$, we can define the induced representation $\pi_C$ on a local neighbourhood $U$ as

$$
[\pi_C(g)f](x) = \rho(\mathrm{h}(g^{-1}, x)^{-1})f(g^{-1}x).
\tag{4}
$$

Here we have assumed that $\mathrm{h}$ is defined at $(g^{-1}, x)$. If it is not, one would need to change to a different section of $G \to G/H$. One may verify, using the composition law for $\mathrm{h}$ (Sec. 2.2), that Eq. 4 does indeed define a representation of $G$. Moreover, one may verify that $\pi_G(g) \circ \Lambda = \Lambda \circ \pi_C(g)$, i.e. they define isomorphic representations.

We can interpret Eq. 4 as follows. To transform a field, we move the fiber at $g^{-1}x$ to $x$, and we apply a transformation to the fiber itself using $\rho$. This is visualized in Fig. 5 for a planar vector field. Some other examples include an RGB image ($\rho(h) = I_3$), a field of wind directions on earth ($\rho(h)$ a $2 \times 2$ rotation matrix), a diffusion tensor MRI image ($\rho(h)$ a representation of $\mathrm{SO}(3)$ acting on 2-tensors), a regular G-CNN on $\mathbb{Z}^3$ [14, 15] ($\rho$ a regular representation of $H$).

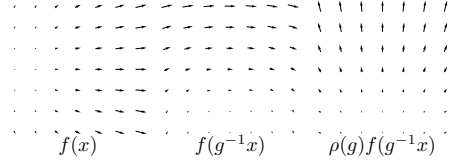

Figure 5: The rotation of a planar vector field in two steps: moving each vector to its new position without changing its orientation, and then rotating the vectors.

## 3 Equivariant Maps and Convolutions

Each feature space in a G-CNN is defined as the space of sections of some associated vector bundle, defined by a choice of base $G/H$ and representation $\rho$ of $H$ that describes how the fibers transform. A layer in a G-CNN is a map between these feature spaces that is equivariant to the induced representations acting on them. In this section we will show that equivariant *linear* maps can always be written as a convolution-like operation using an equivariant kernel. We will first derive this result for the induced representation realized in the space $\mathcal{I}_G$ of Mackey functions, and then convert the result to local sections of the associated vector bundle in Section 3.2. We will assume that $G$ is locally compact and unimodular.

Consider adjacent feature spaces $i = 1, 2$ with a representation $(\rho_i, V_i)$ of $H_i \leq G$. Let $\pi_i = \mathrm{Ind}_{H_i}^G \rho_i$ be the representation acting on $\mathcal{I}_G^i$. A bounded linear operator $\mathcal{I}_G^1 \to \mathcal{I}_G^2$ can be written as

$$
[\kappa \cdot f](g) = \int_G \kappa(g, g')f(g')dg',
\tag{5}
$$

using a two-argument linear operator-valued kernel $\kappa : G \times G \to \mathrm{Hom}(V_1, V_2)$, where $\mathrm{Hom}(V_1, V_2)$ denotes the space of linear maps $V_1 \to V_2$. Choosing bases, we get a matrix-valued kernel.

We are interested in the space of equivariant linear maps between induced representations, defined as $\mathcal{H} = \mathrm{Hom}_G(\mathcal{I}^1, \mathcal{I}^2) = \{\Phi \in \mathrm{Hom}(\mathcal{I}^1, \mathcal{I}^2) \,|\, \Phi\pi_1(g) = \pi_2(g)\Phi, \, \forall g \in G\}$. In order for Eq. 5 to define an equivariant map $\Phi \in \mathcal{H}$, the kernel $\kappa$ must satisfy a constraint. By (partially) resolving this constraint, we will show that Eq. 5 can always be written as a cross-correlation[1]

**Theorem 3.1.** *(convolution is all you need) An equivariant map $\Phi \in \mathcal{H}$ can always be written as a convolution-like integral.*

*Proof.* Since we are only interested in equivariant maps, we get a constraint on $\kappa$. For all $u, g \in G$:

$$
\begin{aligned}
& & [\kappa \cdot [\pi_1(u)f]](g) &= [\pi_2(u)[\kappa \cdot f]](g) \\
&\Leftrightarrow & \int_G \kappa(g, g')f(u^{-1}g')dg' &= \int_G \kappa(u^{-1}g, g')f(g')dg' \\
&\Leftrightarrow & \int_G \kappa(g, ug')f(g')dg' &= \int_G \kappa(u^{-1}g, g')f(g')dg' \\
&\Leftrightarrow & \kappa(g, ug') &= \kappa(u^{-1}g, g') \\
&\Leftrightarrow & \kappa(ug, ug') &= \kappa(g, g')
\end{aligned}
\tag{6}
$$

Hence, without loss of generality, we can define the two-argument kernel $\kappa(\cdot, \cdot)$ in terms of a one-argument kernel: $\kappa(g^{-1}g') \equiv \kappa(e, g^{-1}g') = \kappa(ge, gg^{-1}g') = \kappa(g, g')$.

The application of $\kappa$ to $f$ thus reduces to a cross-correlation:

$$
[\kappa \cdot f](g) = \int_G \kappa(g, g')f(g')dg' = \int_G \kappa(g^{-1}g')f(g')dg' = [\kappa \star f](g).
\tag{7}
$$

$\square$

## 3.1 The Space of Equivariant Kernels

The constraint Eq. 6 implies a constraint on the one-argument kernel $\kappa$. The space of admissible kernels is in one-to-one correspondence with the space of equivariant maps. Here we give three different characterizations of this space of kernels. Detailed proofs can be found in Appendix B.

**Theorem 3.2.** *$\mathcal{H}$ is isomorphic to the space of bi-equivariant kernels on $G$, defined as:*

$$
\begin{aligned}
\mathcal{K}_G = \{\kappa : G \to \mathrm{Hom}(V_1, V_2) \,|\, &\kappa(h_2 g h_1) = \rho_2(h_2)\kappa(g)\rho_1(h_1), \\
&\forall g \in G, h_1 \in H_1, h_2 \in H_2\}.
\end{aligned}
\tag{8}
$$

*Proof.* It is easily verified (see supp. mat.) that right equivariance follows from the fact that $f \in \mathcal{I}_G^1$ is a Mackey function, and left equivariance follows from the requirement that $\kappa \star f \in \mathcal{I}_G^2$ should be a Mackey function. The isomorphism is given by $\Gamma_G : \mathcal{K}_G \to \mathcal{H}$ defined as $[\Gamma_G \kappa]f = \kappa \star f$. $\square$

The analogous result for the two argument kernel is that $\kappa(gh_2, g'h_1)$ should be equal to $\rho_2(h_2^{-1})\kappa(g, g')\rho_1(h_1)$ for $g, g' \in G, h_1 \in H_1, h_2 \in H_2$. This has the following interesting interpretation: $\kappa$ is a section of a certain associated bundle. We define a right-action of $H_1 \times H_2$ on $G \times G$ by setting $(g, g') \cdot (h_1, h_2) = (gh_1, g'h_2)$ and a representation $\rho_{12}$ of $H_1 \times H_2$ on $\mathrm{Hom}(V_1, V_2)$ by setting $\rho_{12}(h_1, h_2)\Psi = \rho_2(h_2)\Psi\rho_1(h_1^{-1})$ for $\Psi \in \mathrm{Hom}(V_1, V_2)$. Then the constraint on $\kappa(\cdot, \cdot)$ can be written as $\kappa((g, g') \cdot (h_1, h_2)) = \rho_{12}((h_1, h_2)^{-1})\kappa((g, g'))$. We recognize this as the condition of being a Mackey function (Eq. 1) for the bundle $(G \times G) \times_{\rho_{12}} \mathrm{Hom}(V_1, V_2)$.

There is another another way to characterize the space of equivariant kernels:

**Theorem 3.3.** *$\mathcal{H}$ is isomorphic to the space of left-equivariant kernels on $G/H_1$, defined as:*

$$
\begin{aligned}
\mathcal{K}_C = \{\overleftarrow{\kappa} : G/H_1 \to \mathrm{Hom}(V_1, V_2) \,|\, &\overleftarrow{\kappa}(h_2 x) = \rho_2(h_2)\overleftarrow{\kappa}(x)\rho_1(\mathrm{h}_1(x, h_2)^{-1}), \\
&\forall h_2 \in H_2, x \in G/H_1\}
\end{aligned}
\tag{9}
$$

*Proof.* using the decomposition $g = s(gH_1)h_1(g)$ (see Appendix A), we can define

$$\kappa(g) = \kappa(s(gH_1)h_1(g)) = \kappa(s(gH_1))\rho_1(h_1(g)) \equiv \overleftarrow{\kappa}(gH_1)\rho_1(h_1(g)), \tag{10}$$

This defines the lifting isomorphism for kernels, $\Lambda_{\mathcal{K}} : \mathcal{K}_C \to \mathcal{K}_G$. It is easy to verify that when defined in this way, $\kappa$ satisfies right $H_1$-equivariance.

We still have the left $H_2$-equivariance constraint from Eq. 8, which translates to $\overleftarrow{\kappa}$ as follows (details in supp. mat.). For $g \in G$, $h_2 \in H_2$ and $x \in G/H_1$,

$$\kappa(h_2 g) = \rho_2(h_2)\kappa(g) \Leftrightarrow \overleftarrow{\kappa}(h_2 x) = \rho_2(h_2)\overleftarrow{\kappa}(x)\rho_1(h_1(x, h_2)^{-1}). \tag{11}$$

$\square$

**Theorem 3.4.** $\mathcal{H}$ *is isomorphic to the space of* $H_2^{\gamma(x)H_1}$*-equivariant kernels on* $H_2\backslash G/H_1$:

$$\mathcal{K}_D = \{\bar{\kappa} : H_2\backslash G/H_1 \to \mathrm{Hom}(V_1, V_2) \,|\, \bar{\kappa}(x) = \rho_2(h)\bar{\kappa}(x)\rho_1^x(h)^{-1},$$
$$\forall x \in H_2\backslash G/H_1, h \in H_2^{\gamma(x)H_1}\}, \tag{12}$$

*Where* $\gamma : H_2\backslash G/H_1 \to G$ *is a choice of double coset representatives, and* $\rho_1^x$ *is a representation of the stabilizer* $H_2^{\gamma(x)H_1} = \{h \in H_2 \,|\, h\gamma(x)H_1 = \gamma(x)H_1\} \leq H_1$, *defined as*

$$\rho_1^x(h) = \rho_1(h_1(\gamma(x)H_1, h)) = \rho_1(\gamma(x)^{-1}h\gamma(x)), \tag{13}$$

*Proof.* In supplementary material. For examples, see Section 6. $\square$

## 3.2 Local Sections on $G/H$

We have seen that an equivariant map between spaces of Mackey functions can always be realized as a cross-correlation on $G$, and we have studied the properties of the kernel, which can be encoded as a kernel on $G$ or $G/H_1$ or $H_2\backslash G/H_1$, subject to the appropriate constraints. When implementing a G-CNN, it would be wasteful to use a Mackey function on $G$, so we need to understand what it means for fields realized by local functions $f : U \to V$ for $U \subseteq G/H_1$. This is done by sandwiching the cross-correlation $\kappa \star : \mathcal{I}_G^1 \to \mathcal{I}_G^2$ with the lifting isomorphisms $\Lambda_i : \mathcal{I}_C^i \to \mathcal{I}_G^i$.

$$[\Lambda_2^{-1}[\kappa \star [\Lambda_1 f]]](x) = \int_G \kappa(s_2(x)^{-1}s_1(y))f(y)dy$$
$$= \int_{G/H_1} \overleftarrow{\kappa}(s_2(x)^{-1}y)\rho_1(h_1(s_2(x)^{-1}s_1(y)))f(y)dy \tag{14}$$

Which we refer to as the $\rho_1$-twisted cross-correlation on $G/H_1$. We note that for semidirect product groups, the $\rho_1$ factor disappears and we are left with a standard cross-correlation on $G/H_1$ with an equivariant kernel $\overleftarrow{\kappa} \in \mathcal{K}_C$. We note the similarity of this expression to gauge equivariant convolution as defined in [11].

## 3.3 Equivariant Nonlinearities

The network as a whole is equivariant if all of its layers are equivariant. So our theory would not be complete without a discussion of equivariant nonlinearities and other kinds of layers. In a regular G-CNN [1], $\rho$ is the regular representation of $H$, which means that it can be realized by permutation matrices. Since permutations and pointwise nonlinearities commute, any such nonlinearity can be used. For other kinds of representations $\rho$, special equivariant nonlinearities must be used. Some choices include norm nonlinearities [3] for unitary representations, tensor product nonlinearities [8], or gated nonlinearities where a scalar field is normalized by a sigmoid and then multiplied by another field [6]. Other constructions, such as batchnorm and ResNets, can also be made equivariant [1, 2]. A comprehensive overview and comparison over equivariant nonlinearities can be found in [7].

# 4 Implementation

Several different approaches to implementing group equivariant CNNs have been proposed in the literature. The implementation details thereby depend on the specific choice of symmetry group $G$, the homogeneous space $G/H$, its discretization and the representation $\rho$. In any case, since the equivariance constraints on convolution kernels are linear, the space of $H$-equivariant kernels is a linear subspace of the unrestricted kernel space. This implies that it is sufficient to solve for a basis of $H$-equivariant kernels, in terms of which any equivariant kernel can be expanded using learned weights.

A case of high practical importance are equivariant CNNs on Euclidean spaces $\mathbb{R}^d$. Implementations mostly operate on discrete pixel grids. In this case, the steerable kernel basis is typically pre-sampled on a small grid, linearly combined during the forward pass, and then used in a standard convolution routine. The sampling procedure requires particular attention since it might introduce aliasing artifacts [4, 6]. A more in depth discussion of an implementation of equivariant CNNs, operating on Euclidean pixel grids, is provided in [7]. Alternatively to processing signals on a pixel grid, signals on Euclidean spaces might be sampled on an irregular point cloud. In this case the steerable kernel space is typically implemented as an analytical function, which is subsequently sampled on the cloud [5].

Implementations of spherical CNNs depend on the choice of signal representation as well. In [10], the authors choose a spectral approach to represent the signal and kernels in Fourier space. The equivariant convolution is performed by exploiting the Fourier theorem. Other approaches define the convolution spatially. In these cases, some grid on the sphere is chosen on which the signal is sampled. As in the Euclidean case, the convolution is performed by matching the signal with a $H$-equivariant kernel, which is being expanded in terms of a pre-computed basis.

# 5 Related Work

In Appendix D, we provide a systematic classification of equivariant CNNs on homogeneous spaces, according to the theory presented in this paper. Besides these references, several papers deserve special mention. Most closely related is the work of [12], whose theory is analogous to ours, but only covers scalar fields (corresponding to using a trivial representation $\rho(h) = I$ in our theory). A proper treatment of general fields as we do here is more difficult, as it requires the use of fiber bundles and induced representations. The first use of induced representations and fields in CNNs is [2], and the first CNN on a non-trivial homogeneous space (the Sphere) is [16].

A framework for (non-convolutional) networks equivariant to finite groups was presented by [17], and equivariant set and graph networks are analyzed by [18–21]. Our use of fields (with $\rho$ block-diagonal) can be viewed as a formalization of convolutional capsules [22, 23]. Other related work includes [24–31]. A preliminary version of this paper appeared as [32].

For mathematical background, we recommend [13, 33–37]. The study of induced representations and equivariant maps between them was pioneered by Mackey [38–41], who rigorously proved results essentially similar to the ones in this paper, though presented in a more abstract form that may not be easy to recognize as having relevance to the theory of equivariant CNNs.

# 6 Concrete Examples

## 6.1 The rotation group $\mathrm{SO}(3)$ and spherical CNNs

The group of 3D rotations $\mathrm{SO}(3)$ is a three-dimensional manifold that can be parameterized by ZYZ Euler angles $\alpha \in [0, 2\pi)$, $\beta \in [0, \pi]$ and $\gamma \in [0, 2\pi)$, i.e. $g = Z(\alpha)Y(\beta)Z(\gamma)$, (where $Z$ and $Y$ denote rotations around the Z and Y axes). For this example we choose $H = H_1 = H_2 = \mathrm{SO}(2) = \{Z(\alpha) \,|\, \alpha \in [0, 2\pi)\}$ as the group of rotations around the Z-axis, i.e. the stabilizer subgroup of the north pole of the sphere. A left $H$-coset is then a subset of $\mathrm{SO}(3)$ of the form

$$gH = \{Z(\alpha)Y(\beta)Z(\gamma)Z(\alpha') \,|\, \alpha' \in [0, 2\pi)\} = \{Z(\alpha)Y(\beta)Z(\alpha') \,|\, \alpha' \in [0, 2\pi)\}.$$

Thus, the coset space $G/H$ is the sphere $S^2$, parameterized by spherical coordinates $\alpha$ and $\beta$. As expected, the stabilizer $H_x$ of a point $x \in S^2$ is the set of rotations around the axis through $x$, which is isomorphic to $H = \mathrm{SO}(2)$.

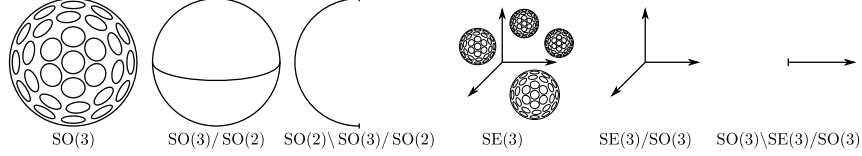

SO(3)     SO(3)/SO(2)   SO(2)\SO(3)/SO(2)    SE(3)     SE(3)/SO(3)   SO(3)\SE(3)/SO(3)

Figure 6: Quotients of SO(3) and SE(3).

What about the double coset space (Appendix A.1)? The orbit of a point $x(\alpha, \beta) \in S^2$ under $H$ is a circle around the $Z$ axis at lattitude $\beta$, so the double coset space $H\backslash G/H$, which indexes these orbits, is the segment $[0, \pi)$ (see Fig. 6).

The section $s : G/H \to G$ may be defined (almost everywhere) as $s(\alpha, \beta) = Z(\alpha)Y(\beta) \in SO(3)$, and $\gamma(\beta) = Y(\beta) \in SO(3)$. Then the stabilizer $H_2^{\gamma(\beta)H_1}$ for $\beta \in H\backslash G/H$ is the set of Z-axis rotations that leave the point $\gamma(\beta)H_1 = (0, \beta) \in S^2$ invariant. For the north and south pole ($\beta = 0$ or $\beta = \pi$), this stabilizer is all of $H = SO(2)$, but for other points it is the trivial subgroup $\{e\}$.

Thus, according to Theorem 3.4, the equivariant kernels are matrix-valued functions on the segment $[0, \pi)$, that are mostly unconstrained (except at the poles). As functions on $G/H_1$ (Theorem 3.3), they are matrix-valued functions satisfying $\overleftarrow{\kappa}(rx) = \rho_2(r)\overleftarrow{\kappa}(x)\rho_1(\mathrm{h}_1(x, r)^{-1})$ for $r \in SO(2)$ and $x \in S^2$. This says that as a function on the sphere $\overleftarrow{\kappa}$ is determined on SO(2)-orbits $\{rx \mid r \in SO(2)\}$ (lattitudinal circles around the Z axis) by its value on one point of the orbit. Indeed, if $\rho(h) = 1$ is the trivial representation, we see that $\overleftarrow{\kappa}$ is constant on these orbits, in agreement with [42] who use isotropic filters. For $\rho_2$ a regular representation of $SO(2)$, we recover the non-isotropic method of [10]. For segmentation tasks, one can use a trivial representation for $\rho_2$ in the output layer to obtain a scalar feature map on $S^2$, analogous to [43]. Other choices, such as $\rho$ the standard 2D representation of $SO(2)$, would make it possible to build spherical CNNs that can process vector fields, but this has not been done yet.

## 6.2 The roto-translation group SE(3) and 3D Steerable CNNs

The group of rigid body motions SE(3) is a 6D manifold $\mathbb{R}^3 \rtimes SO(3)$. We choose $H = H_1 = H_2 = SO(3)$ (rotations around the origin). A left $H$-coset is a set of the form $gH = trH = \{trr' \mid r' \in SO(3)\} = \{tr \mid r \in SO(3)\}$ where $t$ is the translation component of $g$. Thus, the coset space $G/H$ is $\mathbb{R}^3$. The stabilizer $H_x$ of a point $x \in \mathbb{R}^3$ is the set of rotations around $x$, which is isomorphic to SO(3). The orbit of a point $x \in \mathbb{R}^3$ is a spherical shell of radius $\|x\|$, so the double coset space $H\backslash G/H$, which indexes these orbits, is the set of radii $[0, \infty)$.

Since SE(3) is a trivial principal SO(3) bundle, we can choose a global section $s : G/H \to G$ by taking $s(x)$ to be the translation by $x$. As double coset representatives we can choose $\gamma(\|x\|)$ to be the translation by $(0, 0, \|x\|)$. Then the stabilizer $H_2^{\gamma(\|x\|)H_1}$ for $\|x\| \in H\backslash G/H$ is the set of rotations around Z, i.e. SO(2), except for $\|x\| = 0$, where it is SO(3).

For any representations $\rho_1, \rho_2$, the equivariant maps between sections of the associated vector bundle are given by convolutions with matrix-valued kernels on $\mathbb{R}^3$ that satisfy $\overleftarrow{\kappa}(rx) = \rho_2(r)\overleftarrow{\kappa}(x)\rho_1(r^{-1})$ for $r \in SO(3)$ and $x \in \mathbb{R}^3$. This follows from Theorem 3.3 with the simplification $\mathrm{h}_1(x, r) = r$ for all $r \in H$, because SE(3) is a semidirect product (Appendix A.2). Alternatively, we can define $\overleftarrow{\kappa}$ in terms of $\bar{\kappa}$, which is a kernel on $H\backslash G/H = [0, \infty)$ satisfying $\bar{\kappa}(x) = \rho_2(r)\bar{\kappa}(x)\rho_1(r)$ for $r \in SO(2)$ and $x \in [0, \infty)$. This is in agreement with the results obtained by [6].

## 7 Conclusion

In this paper we have developed a general theory of equivariant convolutional networks on homogeneous spaces using the formalism of fiber bundles and fields. Field theories are the de facto standard formalism for modern physical theories, and this paper shows that the same formalism can elegantly describe the de facto standard learning machine: the convolutional network and its generalizations. By connecting this very successful class of networks to modern theories in mathematics and physics, our theory provides many opportunities for the development of new theoretical insights about deep learning, and the development of new equivariant network architectures.

## Footnotes

*Qualcomm AI Research is an initiative of Qualcomm Technologies, Inc.

[1]As in most of the CNN literature, we will not be precise about distinguishing convolution and correlation.

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
