[Supplementary Material]

# Supplementary Material:
# A General Theory of Equivariant CNNs on Homogeneous Spaces

**Taco S. Cohen**
Qualcomm AI Research
Qualcomm Technologies Netherlands B.V.
`tacos@qti.qualcomm.com`

**Mario Geiger**
PCSL Research Group
EPFL
`mario.geiger@epfl.ch`

**Maurice Weiler**
QUVA Lab
U. of Amsterdam
`m.weiler@uva.nl`

## A   General facts about Groups and Quotients

Let $G$ be a group and $H$ a subgroup of $G$. A left coset of $H$ in $G$ is a set $gH = \{gh \mid h \in H\}$ for $g \in G$. The cosets form a partition of $G$. The set of all cosets is called the quotient space or coset space, and is denoted $G/H$. There is a canonical projection $p : G \to G/H$ that assigns to each element $g$ the coset it is in. This can be written as $p(g) = gH$. Fig. 1 provides an illustration for the group of symmetries of a triangle, and the subgroup $H$ of reflections.

The quotient space carries a left action of $G$, which we denote with $ux$ for $u \in G$ and $x \in G/H$. This works fine because this action is associative with the group operation:

$$u(gH) = (ug)H. \tag{1}$$

for $u, g \in G$. One may verify that this action is well defined, i.e. does not depend on the particular coset representative $g$. Furthermore, the action is transitive, meaning that we can reach any coset from any other coset by transforming it with an appropriate $u \in G$. A space like $G/H$ on which $G$ acts transitively is called a homogeneous space for $G$. Indeed, any homogeneous space is isomorphic to some quotient space $G/H$.

A section of $p$ is a map $s : G/H \to G$ such that $p \circ s = \mathrm{id}_{G/H}$. We can think of $s$ as choosing a coset representative for each coset, i.e. $s(x) \in x$. In general, although $p$ is unique, $s$ is not; there can be many ways to choose coset representatives. However, the constructions we consider will always be independent of the particular choice of section.

Although it is not strictly necessary, we will assume that $s$ maps the coset $H = eH$ of the identity to the identity $e \in G$:

$$s(H) = e \tag{2}$$

We can always do this, for given a section $s'$ with $s'(H) = h \neq e$, we can define the section $s(x) = h^{-1}s'(hx)$ so that $s(H) = h^{-1}s'(hH) = h^{-1}s'(H) = h^{-1}h = e$. This is indeed a section, for $p(s(x)) = p(h^{-1}s'(hx)) = h^{-1}p(s'(hx)) = h^{-1}hx = x$ (where we used Eq. 1 which can be rewritten as $up(g) = p(ug)$).

One useful rule of calculation is

$$(gs(x))H = g(s(x)H) = gx = s(gx)H, \tag{3}$$

for $g \in G$ and $x \in G/H$. The projection onto $H$ is necessary, for in general $gs(x) \neq s(gx)$. These two terms are however related, through a function $\mathrm{h} : G/H \times G \to H$, defined as follows:

$$gs(x) = s(gx)\mathrm{h}(x, g) \tag{4}$$

That is,

$$\boxed{\mathrm{h}(x, g) = s(gx)^{-1}gs(x)}. \tag{5}$$

Figure 1: A Cayley diagram of the group $D3$ of symmetries of a triangle. The group is generated by rotations $r$ and flips $f$. The elements of the group are indicated by hexagons. The red arrows correspond to right multiplication by $r$, while the blue lines correspond to right multiplication by $f$. Cosets of the group of flips ($H = \{e, f\}$) are shaded in gray. As always, the cosets partition the group. As coset representatives, we choose $s(H) = e$, $s(rH) = r$, and $s(r^2H) = r^2f$. The difference between $s(rx)$ and $rs(x)$ is indicated. For this choice of section, we must set $\mathrm{h}(x, r) = \mathrm{h}(rH, r) = f$, so that $s(rx)\mathrm{h}(x, r) = (r^2f)(f) = r^2 = rs(x)$.

We can think of $\mathrm{h}(x, g)$ as the element of $H$ that we can apply to $s(gx)$ (on the right) to get $gs(x)$. The $\mathrm{h}$ function will play an important role in the definition of the induced representation, and is illustrated in Fig. 1.

From the fiber bundle perspective, we can interpret Eq. 5 as follows. The group $G$ can be viewed as a principal bundle with base space $G/H$ and fibers $gH$. If we apply $g$ to the coset representative $s(x)$, we move to a different coset, namely the one represented by $s(gx)$ (representing a different point in the base space). Additionally, the fiber is twisted by the right action of $\mathrm{h}(x, g)$. That is, $\mathrm{h}(x, g)$ moves $s(gx)$ to another element in its coset, namely to $gs(x)$.

The following composition rule for $\mathrm{h}$ is very useful in derivations:

$$
\begin{aligned}
\mathrm{h}(x, g_1 g_2) &= s(g_1 g_2 x)^{-1} g_1 g_2 s(x) \\
&= [s(g_1 g_2 x)^{-1} g_1 s(g_2 x)][s(g_2 x)^{-1} g_2 s(x)] \\
&= \mathrm{h}(g_2 x, g_1)\mathrm{h}(x, g_2)
\end{aligned}
\tag{6}
$$

For elements $h \in H$, we find:

$$
\mathrm{h}(H, h) = s(H)^{-1} h s(H) = h.
\tag{7}
$$

Also, for any coset $x$,

$$
\mathrm{h}(H, s(x)) = s(s(x)H)^{-1} s(x) s(H) = s(H) = e.
\tag{8}
$$

Using Eq. 6 and 8, this yields,

$$
\mathrm{h}(H, s(x)h) = \mathrm{h}(hH, s(x))\mathrm{h}(H, h) = h,
\tag{9}
$$

for any $h \in H$ and $x \in G/H$.

For $x = H$, Eq. 5 specializes to:

$$
g = gs(H) = s(gH)\mathrm{h}(H, g) \equiv s(gH)\mathrm{h}(g),
\tag{10}
$$

where we defined

$$
\boxed{\mathrm{h}(g) = \mathrm{h}(H, g) = s(gH)^{-1} g}
\tag{11}
$$

This shows that we can always factorize $g$ *uniquely* into a part $s(gH)$ that represents the coset of $g$, and a part $\mathrm{h}(g) \in H$ that tells us where $g$ is within the coset:

$$
g = s(gH)\mathrm{h}(g)
\tag{12}
$$

A useful property of $\mathrm{h}(g)$ is that for any $h \in H$,

$$
\mathrm{h}(gh) = s(ghH)^{-1} gh = s(gH)^{-1} gh = \mathrm{h}(g)h.
\tag{13}
$$

It is also easy to see that

$$\mathrm{h}(s(x)) = e. \tag{14}$$

When dealing with different subgroups $H_1$ and $H_2$ of $G$ (associated with the input and output space of an intertwiner), we will write $h_i$ for an element of $H_i$, $s_i : G/H_i \to G$, for the corresponding section, and $\mathrm{h}_i : G/H_i \times G \to H_i$ for the h-function (for $i = 1, 2$).

## A.1 Double cosets

A $(H_2, H_1)$-double coset is a set of the form $H_2 g H_1$ for $H_2, H_1$ subgroups of $G$. The space of $(H_2, H_1)$-double cosets is called $H_2 \backslash G / H_1 \equiv \{H_2 g H_1 \,|\, g \in G\}$. As with left cosets, we assume a section $\gamma : H_2 \backslash G / H_1 \to G$ is given, satisfying $\gamma(H_2 g H_1) \in H_2 g H_1$.

The double coset space $H_2 \backslash G / H_1$ can be understood as the space of $H_2$-orbits in $G/H_1$, that is, $H_2 \backslash G / H_1 = \{H_2 x | x \in G/H_1\}$. Note that although $G$ acts transitively on $G/H_1$ (meaning that there is only one $G$-orbit in $G/H_1$), the subgroup $H_2$ does not. Hence, the space $G/H_1$ splits into a number of disjoint orbits $H_2 x$ (for $x = g H_1 \in G/H_1$), and these are precisely the double cosets $H_2 g H_1$.

Of course, $H_2$ *does* act transitively within a single orbit $H_2 x$, sending $x \mapsto h_2 x$ (both of which are in $H_2 x$, for $x \in G/H_1$). In general this action is not necessarily fixed point free which means that there may exist some $h_2 \in H_2$ which map the left cosets to themselves. These are exactly the elements in the stabilizer of $x = g H_1$, given by

$$
\begin{aligned}
H_2^x &= \{h \in H_2 \,|\, hx = x\} \\
&= \{h \in H_2 \,|\, h s_1(x) H_1 = s_1(x) H_1\} \\
&= \{h \in H_2 \,|\, h s_1(x) \in s_1(x) H_1\} \\
&= \{h \in H_2 \,|\, h \in s_1(x) H_1 s_1(x)^{-1}\} \\
&= s_1(x) H_1 s_1(x)^{-1} \cap H_2.
\end{aligned}
\tag{15}
$$

Clearly, $H_2^x$ is a subgroup of $H_2$. Furthermore, $H_2^x$ is conjugate to (and hence isomorphic to) the subgroup $s_1(x)^{-1} H_2^x s_1(x) = H_1 \cap s_1(x)^{-1} H_2 s_1(x)$, which is a subgroup of $H_1$.

For double cosets $x \in H_2 \backslash G / H_1$, we will overload the notation to $H_2^x \equiv H_2^{\gamma(x) H_1}$. Like the coset stabilizer, this double coset stabilizer can be expressed as

$$H_2^x = \gamma(x) H_1 \gamma(x)^{-1} \cap H_2 \tag{16}$$

## A.2 Semidirect products

For a semidirect product group $G$, such as $\mathrm{SE}(2) = \mathbb{R}^2 \rtimes \mathrm{SO}(2)$, some things simplify. Let $G = N \rtimes H$ where $H \leq G$ is a subgroup, $N \leq G$ is a normal subgroup and $N \cap H = \{e\}$. For every $g \in G$ there is a unique way of decomposing it into $nh$ where $n \in N$ and $h \in H$. Thus, the left $H$ coset of $g \in G$ depends only on the $N$ part of $g$:

$$gH = nhH = nH \tag{17}$$

It follows that for a semidirect product group, we can define the section so that it always outputs an element of $N \subseteq G$, instead of a general element of $G$. Specifically, we can set $s(gH) = s(nhH) = s(nH) = n$. It follows that $s(nx) = ns(x) \quad \forall n \in N, x \in G/H$. This allow us to simplify expressions involving h:

$$
\begin{aligned}
\mathrm{h}(x, g) &= s(gx)^{-1} g s(x) \\
&= s(gs(x)H)^{-1} g s(x) \\
&= s(\underbrace{gs(x)g^{-1}}_{\in N} gH)^{-1} g s(x) \\
&= \left(gs(x)g^{-1} s(gH)\right)^{-1} g s(x) \\
&= s(gH)^{-1} g \\
&= \mathrm{h}(g)
\end{aligned}
\tag{18}
$$

### A.3 Haar measure

When we integrate over a group $G$, we will use the Haar measure, which is the essentially unique measure $dg$ that is invariant in the following sense:

$$\int_G f(g)dg = \int_G f(ug)dg \quad \forall u \in G. \tag{19}$$

Such measures always exist for locally compact groups, thus covering most cases of interest [Folland, 1995]. For discrete groups, the Haar measure is the counting measure, and integration can be understood as a discrete sum.

We can integrate over $G/H$ by using an integral over $G$,

$$\int_{G/H} f(x)dx = \int_G f(gH)dg. \tag{20}$$

# B Proofs

## B.1 Bi-equivariance of one-argument kernels on $G$

### B.1.1 Left equivariance of $\kappa$

We want the result $\kappa \star f$ (or $\kappa \cdot f$) to live in $\mathcal{I}_G^2$, which means that this function has to satisfy the Mackey condition,

$$
\begin{aligned}
[\kappa \star f](gh_2) &= \rho_2(h_2^{-1})[\kappa \star f](g) \\
\Leftrightarrow \quad \int_G \kappa((gh_2)^{-1}g')f(g')dg' &= \rho_2(h_2^{-1})\int_G \kappa(g^{-1}g')f(g')dg' \\
\Leftrightarrow \quad \kappa(h_2^{-1}g^{-1}g') &= \rho_2(h_2^{-1})\kappa(g^{-1}g') \\
\Leftrightarrow \quad \kappa(h_2 g) &= \rho_2(h_2)\kappa(g)
\end{aligned}
\tag{21}
$$

for all $h_2 \in H_2$ and $g \in G$.

### B.1.2 Right equivariance of $\kappa$

The fact that $f \in \mathcal{I}_G^1$ satisfies the Mackey condition ($f(gh) = \rho_1(h)f(g)$ for $h \in H_1$) implies a symmetry in the correlation $\kappa \star f$. That is, if we apply a right-$H_1$-shift to the kernel, i.e. $[R_h\kappa](g) = \kappa(gh)$, we find that

$$
\begin{aligned}
[[R_h\kappa] \star f](g) &= \int_G \kappa(g^{-1}uh)f(u)du \\
&= \int_G \kappa(g^{-1}u)f(uh^{-1})du \\
&= \int_G \kappa(g^{-1}u)\rho_1(h)f(u)du.
\end{aligned}
\tag{22}
$$

It follows that we can take (for $h \in H_1$),

$$\kappa(gh) = \kappa(g)\rho_1(h). \tag{23}$$

## B.2 Kernels on $H_2\backslash G/H_1$

We have seen the space $\mathcal{K}_C$ of $H_2$-equivariant kernels on $G/H_1$ appear in our analysis of both $\mathcal{I}_G$ and $\mathcal{I}_C$. Kernels in this space have to satisfy the constraint (for $h \in H_2$):

$$\overleftarrow{\kappa}(hy) = \rho_2(h)\overleftarrow{\kappa}(y)\rho_1(\mathrm{h}_1(y,h)^{-1}) \tag{24}$$

Here we will show that this space is equivalent to the space

$$\boxed{\begin{aligned}
\mathcal{K}_D = \{\bar{\kappa} : H_2\backslash G/H_1 \to \mathrm{Hom}(V_1, V_2) \,|\, \bar{\kappa}(x) = \rho_2(h)\bar{\kappa}(x)\rho_1^x(h)^{-1}, \\
\forall x \in H_2\backslash G/H_1, h \in H_2^{\gamma(x)H_1}\},
\end{aligned}} \tag{25}$$

where we defined the representation $\rho_1^x$ of the stabilizer $H_2^{\gamma(x)H_1}$,

$$\rho_1^x(h) = \rho_1(\mathrm{h}_1(\gamma(x)H_1, h))$$
$$= \rho_1(\gamma(x)^{-1}h\gamma(x)), \tag{26}$$

with the section $\gamma : H_2\backslash G/H_1 \to G$ being defined as in section A.1. To show the equivalence of $\mathcal{K}_C$ and $\mathcal{K}_D$, we define an ismorphism $\Omega_\mathcal{K} : \mathcal{K}_D \to \mathcal{K}_C$. We begin by defining $\Omega_\mathcal{K}^{-1}$:

$$\bar{\kappa}(x) = [\Omega_\mathcal{K}^{-1}\overleftarrow{\kappa}](x) = \overleftarrow{\kappa}(\gamma(x)H_1). \tag{27}$$

We verify that for $\overleftarrow{\kappa} \in \mathcal{K}_C$ we have $\bar{\kappa} \in \mathcal{K}_D$. Let $h \in H_2^{\gamma(x)H_1}$, then

$$\bar{\kappa}(x) = \overleftarrow{\kappa}(\gamma(x)H_1)$$
$$= \overleftarrow{\kappa}(h\gamma(x)H_1)$$
$$= \rho_2(h)\overleftarrow{\kappa}(\gamma(x)H_1)\rho_1(\mathrm{h}_1(\gamma(x)H_1, h))^{-1} \tag{28}$$
$$= \rho_2(h)\bar{\kappa}(x)\rho_1^x(h)^{-1}$$

To define $\Omega_\mathcal{K}$, we use the decomposition $y = h\gamma(H_2y)H_1$ for $y \in G/H_1$ and $h \in H_2$. Note that $h$ may not be unique, because $H_2$ does not in general act freely on $G/H_1$.

$$\overleftarrow{\kappa}(y) = [\Omega_\mathcal{K}\bar{\kappa}](y) = [\Omega_\mathcal{K}\bar{\kappa}](h\gamma(H_2y)H_1) = \rho_2(h)\bar{\kappa}(H_2y)\rho_1(\mathrm{h}_1(\gamma(H_2y)H_1, h))^{-1}. \tag{29}$$

We verify that for $\bar{\kappa} \in \mathcal{K}_D$ we have $\overleftarrow{\kappa} \in \mathcal{K}_C$.

$$\overleftarrow{\kappa}(h'y) = \overleftarrow{\kappa}(h'h\gamma(H_2y)H_1)$$
$$= \rho_2(h'h)\bar{\kappa}(H_2y)\rho_1(\mathrm{h}_1(\gamma(H_2y)H_1, h'h))^{-1}$$
$$= \rho_2(h'h)\bar{\kappa}(H_2y)\rho_1(\mathrm{h}_1(h\gamma(H_2y)H_1, h')\mathrm{h}_1(\gamma(H_2y)H_1, h))^{-1}$$
$$= \rho_2(h')\rho_2(h)\bar{\kappa}(H_2y)\rho_1(\mathrm{h}_1(\gamma(H_2y)H_1, h))^{-1}\rho_1(\mathrm{h}_1(h\gamma(H_2y)H_1, h'))^{-1} \tag{30}$$
$$= \rho_2(h')\rho_2(h)\bar{\kappa}(H_2y)\rho_1(\mathrm{h}_1(\gamma(H_2y)H_1, h))^{-1}\rho_1(\mathrm{h}_1(y, h'))^{-1}$$
$$= \rho_2(h')\overleftarrow{\kappa}(y)\rho_1(\mathrm{h}_1(y, h'))^{-1}$$

We verify that $\Omega_\mathcal{K}$ and $\Omega_\mathcal{K}^{-1}$ are indeed inverses:

$$[\Omega_\mathcal{K}[\Omega_\mathcal{K}^{-1}\overleftarrow{\kappa}]](y) = [\Omega_\mathcal{K}[\Omega_\mathcal{K}^{-1}\overleftarrow{\kappa}]](h\gamma(H_2y)H_1)$$
$$= \rho_2(h)[\Omega_\mathcal{K}^{-1}\overleftarrow{\kappa}](H_2y)\rho_1(\mathrm{h}_1(\gamma(H_2y)H_1, h))^{-1}$$
$$= \rho_2(h)\overleftarrow{\kappa}(\gamma(H_2y)H_1)\rho_1(\mathrm{h}_1(\gamma(H_2y)H_1, h))^{-1} \tag{31}$$
$$= \overleftarrow{\kappa}(h\gamma(H_2y)H_1)$$
$$= \overleftarrow{\kappa}(y).$$

In the other direction,

$$[\Omega_\mathcal{K}^{-1}[\Omega_\mathcal{K}\bar{\kappa}]](x) = [\Omega_\mathcal{K}\bar{\kappa}](\gamma(x)H_1)$$
$$= [\Omega_\mathcal{K}\bar{\kappa}](\gamma(H_2\gamma(x)H_1)H_1)$$
$$= \rho_2(e)\bar{\kappa}(H_2\gamma(x)H_1)\rho_1(\mathrm{h}_1(\gamma(H_2\gamma(x)H_1)H_1, e))^{-1} \tag{32}$$
$$= \bar{\kappa}(x)$$

## C  Limitations of the Theory

The theory presented here is quite general but still has several limitations. Firstly, we only cover fields over homogeneous spaces. Although fields can be defined over more general manifolds, and indeed there has been some effort aimed at defining convolutional networks on general (or Riemannian) manifolds [Bronstein et al., 2017], we restrict our attention to homogeneous spaces because they come naturally equipped with a group action to which the network can be made equivariant. A more general theory would not be able to make use of this additional structure.

For reasons of mathematical elegance and simplicity, the theory idealizes feature maps as fields over a possibly continuous base space, but a computer implementation will usually involve discretizing this space. A similar approach is used in signal processing, where discretization is justified by various sampling theorems and band-limit assumptions. It seems likely that a similar theory can be developed for deep networks, but this has not been done yet.

# D  Classification of Equivariant CNNs

| $G$ | $H$ | $G/H$ | $\rho$ | Reference |
|---|---|---|---|---|
| $\mathbb{Z}^2$ | $\{1\}$ | $\mathbb{Z}^2$ | regular | Lecun 1990 LeCun et al. [1990] |
| $p4, p4m$ | $C_4, D_4$ | $\mathbb{Z}^2$ | regular | Cohen 2016 Cohen and Welling [2016], |
| " | " | " | " | Dieleman 2016 Dieleman et al. [2016] |
| $p4, p4m$ | $C_4, D_4$ | $\mathbb{Z}^2$ | irrep & regular | Cohen 2017 Cohen and Welling [2017] |
| $p6, p6m$ | $C_6, D_6$ | $\mathbb{Z}^2$ | regular | Hoogeboom 2018 Hoogeboom et al. [2018] |
| $\mathbb{Z}^3 \rtimes H$ | $D_4, D_{4h}, O, O_h$ | $\mathbb{Z}^3$ | regular | Winkels 2018 Winkels and Cohen [2018] |
| $\mathbb{Z}^3 \rtimes H$ | $V, T_4, O$ | $\mathbb{Z}^3$ | regular | Worrall 2018 Worrall and Brostow [2018] |
| $\mathbb{R}^2 \rtimes C_N$ | $C_N$ | $\mathbb{R}^2$ | regular | Weiler 2017 Weiler et al. [2018a] |
| " | " | " | " | Zhou 2017 Zhou et al. [2017] |
| " | " | " | " | Bekkers 2018 Bekkers et al. [2018] |
| " | " | " | irrep & regular | Marcos 2017 Marcos et al. [2017] |
| SE(2) | SO(2) | $\mathbb{R}^2$ | irrep | Worrall 2017 Worrall et al. [2017] |
| $\mathbb{R}^2 \rtimes H \leq \mathrm{E}(2)$ | $\mathrm{O}(2), \mathrm{SO}(2), C_N, D_N$ | $\mathbb{R}^2$ | any representation | Weiler 2019 Weiler and Cesa [2019] |
| $\mathbb{R}^2 \rtimes (\mathbb{R}^+, *)$ | $(\mathbb{R}^+, *)$ | $\mathbb{R}^2$ | regular & trivial | Ghosh 2019 Ghosh and Gupta [2019] |
| " | " | " | regular | Worrall 2019 Worrall and Welling [2019] |
| " | " | " | " | Sosnovik 2019 Sosnovik et al. [2019] |
| SE(3) | SO(3) | $\mathbb{R}^3$ | irrep | Kondor 2018 Kondor [2018] |
| " | " | " | irrep & regular | Thomas 2018 Thomas et al. [2018] |
| " | " | " | irrep | Weiler 2018 Weiler et al. [2018b] |
| " | " | " | irrep | Kondor 2018 Kondor et al. [2018] |
| " | " | " | irrep | Anderson 2019 Anderson et al. [2019] |
| SO(3) | SO(2) | $S^2$ | regular | Cohen 2018 Cohen et al. [2018] |
| " | " | " | trivial | Esteves 2018 Esteves et al. [2018] |
| " | " | " | " | Perraudin 2018 Perraudin et al. [2018] |
| " | " | " | irrep | Jiang 2019 Jiang et al. [2019] |
| $G$ | $H$ | $G/H$ | trivial | Kondor 2018 Kondor and Trivedi [2018] |

Table 1: A taxonomy of G-CNNs. Methods are classified by the group $G$ they are equivariant to, the subgroup $H$ that acts on the fibers, the base space $G/H$ to which the fibers are attached (implied by $G$ and $H$), and the type of field $\rho$ (regular, irreducible or trivial).