[Reviews · NeurIPS 2019]

Reviewer 1



*********** update after author feedback ***************** The improvements the authors note sound great and I hope this can improve the impact of the paper significantly. I would give an accept score if I were able to have a look at the new version and be happy with it (as is possible in openreview settings for example). However since improving the presentation usually takes a lot of work and it is not possible for me to verify in which way the improvements have actually been implemented, I will bump it to a 5. I do think readability and clarity is key for impact as written in my review, which is the main reason I gave a much lower score than other reviewers, some of whom have worked on exactly this intersection of algebra and G-CNNs themselves and provided valuable feedback on the content from an expert's perspective. The following comments are based on the reviewer's personal definition of clarity and good quality of presentation: that most of the times when following the paper from start to end it is clear to the reader why each paragraph is written and how it links to the objective of the main results of the paper, here claimed e.g. in the last sentence to be the development of new equivariant network architectures. The paper is one long lead-up of three pages of definitions of mathematical terms and symbols to the theorems in section 6 on equivariant kernels which represent the core results of the paper. In general, I appreciate rigorous frameworks which generalize existing methods, especially if they provide insight and enable the design of an arbitrary new instance that fits in the framework (in this case transformations on arbitrary fields). However that being said, my main concern with this paper is that I'm not sure whether the latter is actually achieved because of how the paper is presented in the current state. They describe G-CNNs in the language of field theory which is nice, but they do not elaborate on the exact impact of this, albeit interesting, achievement. Furthermore given my personal definition above, the quality in presentation is severely lacking and my main criticism about the paper. As an ML conference paper, the presentation is not transparent enough 1. for the average researcher to understand the framework to see for which equivariances they could use it 2. to see how exactly they can now design the convolutional layer in their specific instance. Now the authors could aim for the properly trained and interested mathematicians/physicists only - then, however I'm not sure why pages 3-5 of definitions are at such a prominent place in the paper. It is background knowledge that either the trained reader knows already or won't understand in the condensed way it is presented here. In particular, the definitions are given without the reader knowing why and how to connect it the familiar machine learning models/concepts. If they are experts, then these pages occlude the main take-away of the paper for the connection of these field theoretic concepts to practical G-CNN networks, which is discussed very little on the last page. Since the paper introduces everything starting with symmetry groups (admittedly basic concept) however, it does seems to aim to reach someone not too familiar with group theory. However it is inconceivable to me how they would be able to follow through all the way until induced representations and equivariant kernels in a reasonable amount of time, without actually properly learning the background by working through chapters of a book.

Reviewer 2



Having read the authors' feedback and other reviews, I am increasing my score from 5 to 6. If the paper is accepted, I would ask the authors to dedicate more space to worked examples and to differentiate their work from the existing literature in more detail. -------------------------------------------------------- In terms of novelty over previous works on equivariant CNNs, this paper is a mild step forward, bringing in sections of associated vector bundles for feature spaces and allowing for general representations there. The clarity of this mathematical language is admittedly nice and I think it will help researchers think about general equivariant CNNs in the future. However, Section 8 did not do a sufficient job of clarifying what the new theory does that previous papers could not, in terms of relevant examples. The main theorem is a classification of equivariant linear maps between such feature spaces. The organization of the paper is probably not optimal for NeurIPS, with the body of the paper Sections 2-6 reviewing general mathematical constructions. Some of this material could presumably be relegated to appendices, e.g. the proofs in Section 6, leaving more space for improved discussion.

Reviewer 3



The paper studies the following problem: If we consider a base space that admits a transitive action, and if the feature maps in neural network layers operating on this space are fields, then what is the most general way to write equivariant linear maps between the layers? The first contribution of the paper is to state and prove a theorem that says that such a linear map is a cross-correlation/convolution with a specially constrained kernel, which is called an equivariant kernel. The proof follows in a very straightforward manner from the application of MacKay's theorem aka Frobenius reciprocity, which essentially describes how induction and restriction interact with one another. Turns out that this is precisely the language needed to describe the equivariant networks talked about in this paper (and implicitly in many experimental papers). The proof is elegant and natural, and no details are omitted. Next, in a somewhat abstract manner it is also describes how such constrained kernels will look like. This to me personally is the most useful, as for practitioners, it gives a systematic procedure to derive the right kernel for convolution. This is also useful in different ways -- for example there has been recent work that posits that for continuous groups it is perhaps useful to always operate in Fourier space. To enforce locality we then need an appropriate notion of wavelets. The two approaches are equivalent, but I find the approach presented in the paper more transparent vis a vis jointly enorcing locality and equivariance. Appropriate equivariant non-linearities are also described. Lastly useful examples are given re spherical CNNs, SE(3) steerable CNNs that do a good job in making the discussion a bit more concrete (although still in the abstract space :)

Reviewer 4



This is an "emergency review" therefore it might be shorter than the norm. There is a substantial literature on steerability in the classical image processing domain. Recently, it has become clear that generalizing steerability to the action groups other than SE(2) is important for constructing certain classes of neural networks. Steerability can be described in different ways, some more abstract than others. This paper uses the language of fiber bundles, which is beautiful and enlightening but somewhat abstract. The paper makes no apologies about being abstract. I can understand that to somebody who comes more from the applications side of ML rather than the mathematical side it might be difficult to digest. On the other hand, it also states that "This paper does not contain fundamentally new mathematics (in the sense that a professional mathematician with expertise in the relevant subjects would not be surprised by our results)." I like this honesty. In actual fact, I don't think that either of the above detract from the value of the paper. This paper forms an important bridge between the neural nets literature and certain branches of algebra. I found it very enlightening. I appreciate the effort that the authors have made to show how a wide range of other works fit in their framework. I also think that the exposition is very straight forward. It does not attempt to gloss over or hide any of the underlying mathematical concepts. At the same time, it avoids getting bogged down with mathematical minutiae or a long list of definitions. The authors clearly made an attempt to say things in a way that is "as simple as possible but not simpler." It is quite an achievement to expose all the concepts that they need in 8 pages.

[Author Response · NeurIPS 2019]

We would like to thank the reviewers for their careful study of our paper and helpful suggestions. There is some spread in the review scores, with reasonable arguments on all sides, but we think that we can address all of the issues raised.

All reviewers seem to agree that the paper makes a significant contribution in unifying the rapidly expanding literature on equivariant CNNs into a single coherent mathematical framework / language (this language being foundational to modern theoretical physics as well), showing how existing work fits in this framework, showing how convolutions emerge naturally from the requirement of equivariance, and characterizing the admissible (equivariant) convolution kernels. We are also pleased to see that the reviewers generally appreciate the clarity and elegance enabled by the mathematical machinery used in the paper. Finally, it is agreed that the material is inherently technical, that unification requires a certain amount of abstraction, and that at the same time the prerequisite mathematics is not part of the standard ML background knowledge. The main concern raised by the reviewers centers on this difficulty of exposition.

It is our aim to make the paper accessible to the mathematically oriented ML researcher, as well as to mathematicians and physicists who want to enter ML. The former should be able to grasp the general ideas and intuition (and know where to learn more), while the latter should be able to relate the ideas to things they already know. Thus the lengthy mathematical build up serves the dual purpose of introducing the concepts (for those not familiar) and establishing notation (for those who are). That said, we do agree with R1&R4 that the build-up is too long and starts from too elementary a point. A complete novice will not be able to follow anyway. We have thus decided to move all of Section 2 (groups, homogeneous spaces) to the appendix. This gives us $\sim 1$ page of space to address other concerns (see below).

We will keep sections 3-5 (bundles, fields, induced representations) in the main paper, but change the emphasis from formal definitions to intuition and visualization, with some definitions moved to the appendix. In this way, the ML audience can more easily get the general idea while math and physics folks will know enough from reading the names (bundle, field, etc.). We believe that in this way we can address the concerns of R1 and R4 without giving up on the expository style appreciated by R5 and R7.

Given the space freed up by this refactoring, we have been able to substantially improve the discussion based on reviewer suggestions in the following ways:

1. We added an overview after related work where we provide intuition for the main concepts of the theory (bundles, fields, induced rep, equivariant kernel), and discuss their role in the theory. This provides a roadmap for the paper, and makes it clear to the reader where we are going with the definitions that follow (R1).
2. Explain the concept of a principal and associated bundle, as well as the induced representation via the example of a vector field on the sphere (with new figure).
3. When introducing a concept, we explain more explicitly how it relates to familiar ML concepts such as convolutional feature maps and filter banks (R1).
4. Added a short section to explain how the main theorems inform the implementation of G-CNNs (e.g. by parameterizing the kernel as a linear combination of basis kernels that solve the kernel constraint) (R1).
5. We have expanded the examples and made them more readable. Moreover, we explicitly mention where the theory identifies gaps in the literature (R4), such as vector fields (and other fields) on the sphere or diffusion tensor images. These examples can only be described satisfactorily using bundles (R1, R4), a point we clarify in this section.
6. We expanded the discussion of the theorems and their significance. As mentioned by R5, it is important for practitioners to know that the kernel they use is the most general, and one is not imposing additional constraints (unnecessarily limiting expressivity) beyond what is necessary for equivariance to hold.

In response to the question regarding novelty by R4, we would like to point out that the novelty vis a vis Kondor & Trivedi is that we allow for arbitrary fields (not just scalar). In other words, we cover general steerable CNNs and not just regular G-CNNs. Given our framework, the difference is simple to state and may sound small, but without our framework it is actually not at all obvious how to extend K&T to steerable CNNs, or even that fiber bundles / fields and steerable CNNs are in any way related. Concerning novelty wrt application papers, we note that although it is not the main point of our work, our systematic approach does reveal some gaps in the literature (e.g. a network that can process vector fields on the sphere, which is useful in climate science). It also covers methods yet to be discovered.

Although as mentioned in the paper, we do not introduce fundamentally new mathematics, we believe that the observation that all G-CNNs can be described so cleanly in the language of fiber bundles, which is so fundamental to modern mathematics and physics, is highly significant. We believe that together with the work of Kondor & Trivedi, our paper provides a solid foundation for the theory of G-CNNs on homogeneous spaces, and is likely to have a significant influence on future work in this area. It is true that our paper deviates from the standard ML-paper mold, but believe this should in itself not be a reason for rejection. Finally, given the above-mentioned improvements to structure and clarity inspired by the reviewer comments, we think our paper provides a very readable account of the theory.

[Meta-Review · NeurIPS 2019]

The paper provides a clear mathematical framework for unifying several recent advances on generalizing CNNs to spaces acted on by groups. The authors make it clear that they do not present any new architectures or fundamentally new mathematics. They are also unapologetic about the fact that the paper is abstract. Having said all this, it is clear that equivariant neural nets are here to stay and writing a paper that provides the right language to talk about them is an important contribution. While the present readership of this might paper might be limited, in the future it might become a standard reference and important stepping stone to designing other types of networks.